# Time reversed optical waves by arbitrary vector spatiotemporal field generation

Mickael Mounaix [1,3], Nicolas K. Fontaine[2,3], David T. Neilson[2], Roland Ryf[2], Haoshuo Chen[2], Juan Carlos Alvarado-Zacarias[2] & Joel Carpenter [1✉]

Lossless linear wave propagation is symmetric in time, a principle which can be used to create time reversed waves. Such waves are special "pre-scattered" spatiotemporal fields, which propagate through a complex medium as if observing a scattering process in reverse, entering the medium as a complicated spatiotemporal field and arriving after propagation as a desired target field, such as a spatiotemporal focus. Time reversed waves have previously been demonstrated for relatively low frequency phenomena such as acoustics, water waves and microwaves. Many attempts have been made to extend these techniques into optics. However, the much higher frequencies of optics make for very different requirements. A fully time reversed wave is a volumetric field with arbitrary amplitude, phase and polarisation at every point in space and time. The creation of such fields has not previously been possible in optics. We demonstrate time reversed optical waves with a device capable of independently controlling all of light's classical degrees of freedom simultaneously. Such a class of ultrafast wavefront shaper is capable of generating a sequence of arbitrary 2D spatial/polarisation wavefronts at a bandwidth limited rate of 4.4 THz. This ability to manipulate the full field of an optical beam could be used to control both linear and nonlinear optical phenomena.

[1] School of Information Technology and Electrical Engineering, The University of Queensland, Brisbane, QLD 4072, Australia. [2] Nokia Bell Labs, 791 Holmdel Road, Holmdel, NJ 07722, USA. [3] These authors contributed equally: Mickael Mounaix, Nicolas K. Fontaine. ✉email: j.carpenter@uq.edu.au

In a time reversal experiment[1–6], the spatiotemporal field to be recreated is often a short pulse originating from a small focused spot. After potentially undergoing a complicated scattering process the far-field of this source is recorded by an array of transducers/antennas. The time axis of these signals is then flipped and replayed through the array to regenerate the spatially and temporally focused source. This can be extended to a transfer matrix-based approach[7–12], whereby an array of sources is characterised allowing arbitrary superpositions of those sources to be regenerated. Time reversal processing can be performed either by physically back-propagating signals through the medium, or by numerically back-propagating signals using the conjugate transpose of the transfer matrix measured in the forward direction.

An advantage of a transfer matrix approach is the ability to deliver arbitrary spatiotemporal fields to the target, without first having to physically generate, back-propagate and measure the required input field, meaning experimental error associated with physical field generation is eliminated, and any field can be synthesised from a small number of measurements, not just fields, which have been previously measured.

Low frequency phenomena such as acoustics, water waves and microwaves, are within reach of electrical digitisers and signal generators that can record and generate the required fields directly in the time-domain. When working with broadband sources at optical frequencies, such as femtosecond lasers, the electric field cannot be directly measured or manipulated in the time-domain as it can for acoustics or microwaves. Hence extending time reversal techniques into optics requires a different approach.

For a given target field after propagation through a complex medium, the exact corresponding time reversed wave in the general case is a completely arbitrary function of space, time and polarisation. The scattering process itself need not create waves with any correlations between the spatial, temporal and polarisation degrees of freedom, and therefore any device capable of generating these time reversed fields must be similarly unrestricted. The generation of such fields requires independent control of the impulse response of every spatial and polarisation mode supported by the medium. In this sense, it is the ultimate form of linear wave control as it requires all the classical linear properties of the wave to be controlled independently and simultaneously.

Previous experiments in optics[10,13,14] have demonstrated spatial control[9,15–21], temporal control[22–25] or some limited combination of both[26–33]. These demonstrations have various implementations, however, they all share an inability to simultaneously control two-dimensional (2D) space, time/frequency and polarisation as completely independent degrees of freedom. In previous demonstrations, each property of light does not map to its own spatially separated position on the programmable wavefront control device (typically an spatial light modulator (SLM)), where it can be independently controlled. For example, all spectral components may map to the same position on the SLM[26], or one spatial axis cannot be controlled independently of the spectral axis[27,34,35], and it is also common for polarisation to be neglected entirely. Owing to the high frequencies and bandwidths, the manipulation of ultrashort optical pulses is typically performed in the frequency domain using spectral pulse shapers[36]. In the frequency domain, time reversal is broadband phase conjugation[13,27], and requires the amplitude and phase of every spatial mode to be independently controllable as a function of wavelength. A spectral pulse shaper based on a 2D SLM can control the frequency response of multiple spots in a linear array simultaneously[37]. In such a system, one-axis of the SLM is assigned to the wavelength degrees of freedom and the other axis the one-dimensional (1D) spatial degrees of freedom, consisting of the output linear spot array. However, the transverse spatial properties of a beam are 2D. In Cartesian coordinates for example, the transverse structure of a beam is a function of both $x$ and $y$. Hence there are three dimensions of required control per polarisation, which in the simplest case must be addressed by the 2D surface of the SLM. This dimensional mismatch is an important reason why fully time reversed waves in optics have not previously been demonstrated. Previous work used the two dimensions of the SLM to control the two spatial dimensions[9], 1 spatial dimension and 1 spectral dimension[27,34], or some other partial combination of the spatial and temporal degrees of freedom[13,22,26,29,30,32,35].

In this work as summarised in Fig. 1, we employ a multi-plane light conversion (MPLC) device[38–40] in combination with a polarisation-resolved[41,42] multi-port[37] spectral pulse shaper[36] in order to control all three dimensions (2 space, 1 frequency/time) for both polarisations on a 2D SLM. The MPLC device performs a spatial transformation, which maps a 1D array of 45 Gaussian spots to a 2D set of 45 Hermite-Gaussian (HG) modes[39,40]. This

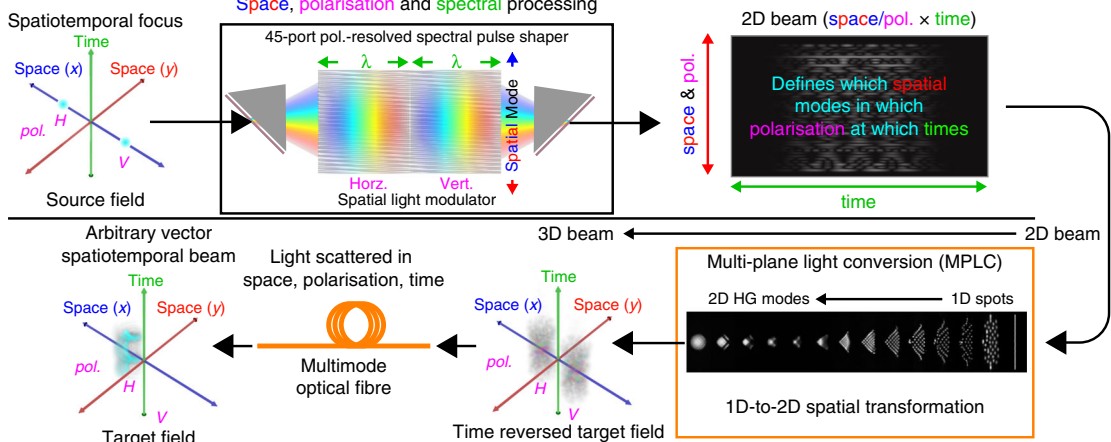

**Fig. 1 Simplified schematic of a device capable of mapping an input vector spatiotemporal field onto an arbitrary vector spatiotemporal output field.** Amplitude, phase, spatial mode, polarisation and spectral/temporal degrees of freedom can all be independently addressed simultaneously through the programming of the spatial light modulator. Ninety spatial/polarisation modes can be independently controlled over 4.4 THz at a resolution of ~15 GHz, making a total of ~26,000 spatiospectral modes. Source, time reversed and target fields have polarisation components illustrated spatially separated for clarity, but are co-located. Optical system animated in detail in second online video (14:00) in Supplementary Note 1.

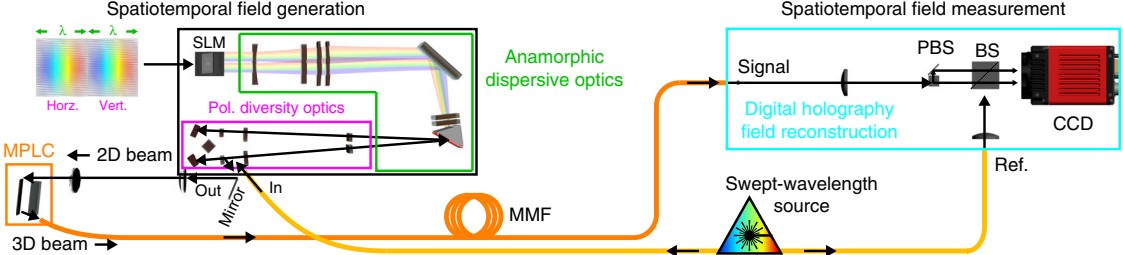

**Fig. 2 Schematic of spatiotemporal field generation and characterisation apparatus.** A polarisation and spatially resolved spectral pulse shaper for generating arbitrary vector spatiotemporal states, in conjunction with a swept-wavelength digital holography system for characterisation. All characterisation and results are measured in the frequency domain.

enables two spatial dimensions of the output beam to be controlled using a single spatial dimension of the SLM, which leaves the other spatial dimension of the SLM for control of the spectral/temporal degree of freedom. The spectral pulse shaper steers light to a set of 45 discrete spot positions in a 1D array, which through the MPLC transformation, excite the corresponding 2D HG modes at the input of the multimode fibre. Addressing the output beam in the HG basis naturally supports applications in both free-space and fibre optics. Compared to spatial bases of 2D spot arrays, the HG basis also has the advantage of no regions of deadspace where light cannot be delivered in the near and far-fields.

## Results

**Experimental setup**. A schematic of the device and the associated characterisation apparatus is summarised in Fig. 2. Detailed explanations and visualisations are available in the Supplementary Information and online videos listed in Supplementary Note 1. The source field enters the device through a single-mode fibre (SMF) in some arbitrary polarisation-dependent temporal state. Inside the device, this source field will be split among many optical paths by the SLM, creating different combinations of spatial modes and delays. This redistribution of light transforms the source into the desired spatiotemporal or spatiospectral state at the output. In these demonstrations, the source corresponds in the time-domain with a bandwidth-limited sinc pulse (4.4 THz of rectangular bandwidth centred at 1551.4 nm), linearly polarised at 45 degrees with respect to the SLM. This source field enters the polarisation-resolved multi-port spectral pulse shaper where it is mapped onto the surface of the SLM (Holoeye PLUTO II) through polarisation diversity optics and anamorphic dispersive optics. Through these optics, the horizontally and vertically polarised components of the beam are separated onto the left and right side of the SLM respectively. Within each beam for each polarisation component, the spectral components are dispersed across the $x$-axis (wide axis) of the SLM. Applying a phase tilt along this spectral axis ($x$-axis) will steer the beam back towards the output array along longer or shorter paths through the grating, creating controllable delay[36]. In Fig. 2, this corresponds with steering in the plane of the page. The device operates between 1533.94 nm and 1569.27 nm, corresponding to 4.4 THz of optical bandwidth. The width of each spectral component on the SLM is ~3 pixels or 15 GHz. Corresponding with a maximum of ~300 spectral modes of control for each of the 90 spatial/polarisation modes. Applying a phase tilt along the $y$-axis (short axis) of the SLM will steer the beam along the 1D array of Gaussian spots at the input to the MPLC[39,43]. In Fig. 2, this corresponds with steering in/out of the plane of the page. Each of the 45 spots in the 1D array is transformed to a particular HG mode through the 14 phase planes of the MPLC device for both polarisation components. In this way, both Cartesian indices ($m,n$) of the HG basis set can be addressed by steering light along the 1D array

using the SLM. By programming more complicated phase masks onto the SLM it is possible to redistribute light among these two axes corresponding to the spatial/polarisation and spectral/temporal degrees of freedom, allowing arbitrary spatial/polarisation modes to be assigned to arbitrary frequencies or delays. The device is attached to a 5 m length of graded-index 50 μm core diameter multimode fibre (MMF)[44], which will be used as a complex medium through which a desired vector spatiotemporal field is to be generated. The fibre supports the same number of modes as the spatiotemporal beam shaper (90 spatial/polarisation modes) with a delay spread of approximately 0.15 ps/m[44]. Owing to modal dispersion, a spatiotemporal state input to the fibre arrives at the other end in a different spatiotemporal state. As the entire delay spread of all spatial and polarisation modes this fibre supports are addressable, no path exists through the fibre, which cannot be time reversed.

A complete linear description of the device and the attached MMF is acquired using swept-wavelength digital holography[39]. This linear description consists of a set of $90 \times 90$ frequency-dependent complex matrices, which maps each of the 90 spatial/polarisation input modes to each of the 90 output modes, in both amplitude and phase as a function of optical frequency. Through these matrices, any spatiotemporal input can be mapped to any spatiotemporal output in both directions. To measure these matrices, each spatial mode in each polarisation is selectively excited at the input of the MMF one-at-a-time by the SLM. Then, as the wavelength of the source is swept, the digital holography system measures the optical field at the output of the fibre, and extracts the amplitude and phase for each output mode as a function of frequency. The desired spatiotemporal state to be generated at the output of the fibre is specified as a wavelength-dependent complex vector, which when back-propagated through the measured matrices yields the corresponding input spatiotemporal state. A phase mask is then calculated to generate this state using a modified Gerchberg-Saxton algorithm and displayed on the SLM. Swept-wavelength digital holography is then performed to characterise the resulting spatiotemporal output state.

In traditional holography, a 2D diffractive element encodes the complex amplitude of a 2D wavefront, which can be recreated by illuminating the element with a spatial reference beam. This new device can be thought of as an extension of this to an extra dimension; a three-dimensional (3D) diffractive element, which when illuminated with a reference pulse in a reference spatial mode, will reconstruct a fully volumetric optical field (2 transverse space and 1 time/longitudinal space). It is a type of reprogrammable space-time hologram[45].

This device is capable of generating types of optical beams, with full control of the spatial, spectral/temporal and polarisation properties and the implementation of various mappings between these properties, for example, arbitrarily polarised focused spots or singularities[46,47] tracing arbitrary trajectories through space and time or frequency[48], or indeed any arbitrary vector spatial fields generated at arbitrary wavelengths or delays. In this way the

device can also be thought of as a kind of ultrafast wavefront shaper, capable of generating a sequence of approximately 90 independent wavefronts at a rate of 4.4 THz. For a traditional continuous wave (CW) beam, the wavefront in one plane specifies the wavefront at all other points in space ahead and behind the wavefront. However, these new beam types have an additional dimension of control. For a fixed point in time, or alternatively a fixed plane in space along the optic axis, like for example the focal plane, the wavefronts ahead and behind can all be independently controlled and unrelated.

Spatial wavefront manipulation and spectral pulse shaping already have broad applicability and the ability to perform both simultaneously by this device could have many applications within linear and nonlinear optics, for example, the control of light propagation through complex media for applications such as imaging, which is analogous to previous demonstrations for lower frequency phenomena. However, optical waves are quite different from acoustics, microwaves and water waves, not only in terms of wavelength, frequency and bandwidth, but also particularly with respect to interaction with matter. Hence, this new type of control in optics could open up many possibilities that are not just generalisations of previous demonstrations for lower frequency phenomena, with applications such as nonlinear microscopy[49], micromachining[50], quantum optics[51], optical trapping[52], nanophotonics and plasmonics[53], optical amplification[54] and other new nonlinear spatiotemporal phenomena, interactions and sources[55–57].

**Experimental results**. Figure 3 contains various examples of spatiospectral and spatiotemporal control, with additional examples available in the Supplementary Information. The measured 3D optical fields (2D space and 1D time/frequency) are plotted as a sequence of 2D fields, as well as volumetric renderings. The plots contain both amplitude and phase information for these fields for both polarisation components. Examples, such as the spatiotemporal foci of Fig. 3b, d, are similar to typical first demonstrations of time reversal in other wave phenomena, which is a use case of practical application in imaging and nonlinear optics. However, most examples were chosen to be illustrative of the system's ability to generate beams with arbitrary control over space, time/frequency and polarisation, rather than any specific use case. All beam types are characterised in the frequency domain. For spatiotemporal beam demonstrations, the presented results are Fourier transformed into the time-domain from the measured frequency-dependent fields measured using swept-wavelength digital holography.

In Fig. 3a the device of Fig. 2 is used as a programmable dispersive element, which can perform arbitrary mappings between wavelength and 2D space/polarisation. The horizontal and vertical polarisations cycle through spatial fields corresponding with letters of the Latin alphabet in forward and reverse alphabetical order respectively. From these measured optical fields of Fig. 3a it is possible to see that the spatial amplitude and phase in both polarisations can be controlled as a function of wavelength. The overall spectral phase is near constant as a function of wavelength and hence all letters arrive at the same delay.

The remaining examples of Fig. 3 all demonstrate spatiotemporal beams. In these examples, the principle of operation is much the same as the spatiospectral control example of Fig. 3a, except the desired spatial/polarisation output states are specified in the time-domain. For temporal control, it is critical that not only are the correct superpositions of spatial/polarisation modes excited at each wavelength, but also the relative phase between the wavelengths must be correct in order to generate the desired temporal features. Figure 3b is a relatively simple spatiotemporal

demonstration; a polarised diffraction limited focus of 230 fs temporal duration (corresponding to the bandwidth of the device, 4.4 THz). As discussed in more detail in the Supplementary Information (Supplementary Fig. 16), at 0 ps delay, the spatial focus is 86% of the peak theoretical intensity achievable using 45 HG modes, and 84% of the theoretically achievable power for a rectangular spectrum of 4.4 THz is delivered to the 0 ps delay. Figure 3c is a similar demonstration of a more complicated spatial field, a vertically polarised "smiley face", which has been generated at a desired delay of 5 ps. Figure 3d in an example of two orthogonally polarised focal spots that are independently tracing out spirals in time across a delay spread of 20 ps. This can be seen from the sequence of 11 measured fields shown in Fig. 3d and from the volumetric rendering of the field to the right of Fig. 3d. This illustrates how the device can be used to trace focused spots along arbitrary trajectories at a rate limited by the optical bandwidth (4.4 THz). This could be used for raster scanning or other kinds of sampling at ultrafast rates. Figure 3e is another spatiotemporal demonstration, similar to Fig. 3d, but which maps more complicated spatial fields representing numerals to different delays across an 8 ps delay spread. Figure 3f is an example that illustrates a spatial and polarisation state change as a function of delay. A radially polarised "clock hand". That is, a spatial field consisting of a linearly polarised line pointing outwards from the centre of the fibre core, which rotates as a function of delay. This kind of control of space, time and polarisation independently could be of use to high-power applications working at high numerical apertures, such as micromachining[50] or optical trapping[52]. The examples of Fig. 3g, h are designed to represent more recognisable 3D objects. Figure 3g, the "arrow of time", corresponds with a light field literally shaped like a 3D arrow. Launching a pulse into the input of the spectral pulse shaper of Fig. 2 would result in a 3D arrow shaped beam, pointed backwards along the time/propagation axis, flying into the charge-coupled device camera of the characterisation apparatus. Similarly, Fig. 3f is a simple 3D light rendering of an Eiffel tower shaped beam flying base-first along the propagation axis. Further experimental results are available in the Supplementary Information, particularly in the online videos, including additional characterisations such as spatially and temporally dependent losses, as well as tests of performance versus number of addressed spatiospectral modes.

The imperfections in the device can be organised into three types. First, there is frequency-dependent loss (Supplementary Fig. 8). The frequency response of the device is not perfectly flat over the full 4.4 THz range, which makes for a slight broadening of the temporal features. Integrating under the average frequency response curve over all 90 spatial/polarisation modes, yields a bandwidth that is 83% compared with a flat 4.4 THz response. This is consistent with the 84% value for the specific example of the spatiotemporal focus presented in Fig. 3b and Supplementary Fig. S16. The second type of imperfection is mode dependent loss (Supplementary Fig. 8). Each of the 90 spatial/polarisation modes do not have the same loss, which leads to a loss of spatial/polarisation detail. This effect averaged over all frequencies and all possible spatial/polarisation target fields yields a spatial/polarisation fidelity of 83%. That is, when maximising the power delivered to the target state, and assuming no additional imperfections that were not present when the transfer matrices were measured, on average 83% of the total output power would be in the target spatial/polarisation state. A thorough investigation of the distribution of spatial fidelities over frequency and target state is provided in Supplementary Fig. 17. Both frequency and spatial/polarisation-dependent losses could at least to some extent be calibrated out by trading off total power delivered to the target state, in favour of a higher proportion of the output power

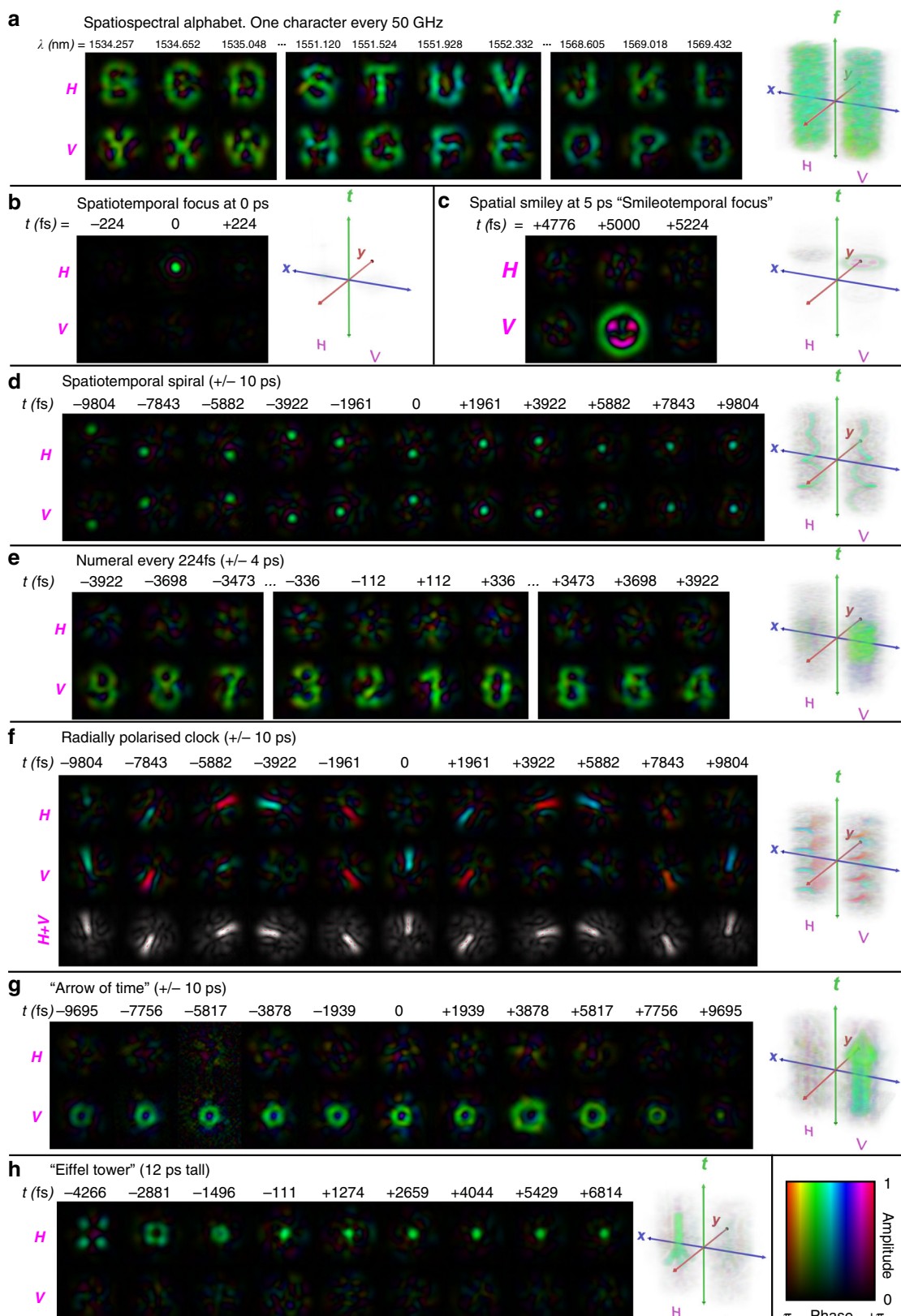

**Fig. 3 Selection of vector spatiospectral and spatiotemporal states measured at the distal end of the multimode optical fibre.** Various examples illustrating control of the spatial amplitude, phase and polarisation of a beam as a function of frequency or time. The examples are shown as a sample sequence of measured optical fields, as well as a volumetric rendering of the field as function of space, time/frequency and polarisation. **a** Spatiotemporal demonstration. **b–h** Spatiotemporal demonstrations are Fourier transforms of the measured optical fields using swept-wavelength digital holography. Further examples and animations are available in the Supplementary Information.

delivered to the target state. In practice, that would mean redistributing and/or attenuating the power of spatial and spectral degrees of freedom to flatten the overall response.

The third imperfection type relates to the accuracy at which arbitrary spatiotemporal/spatiospectral states can be generated. This is largely about the ability of the SLM to accurately represent the required phase masks, in the presence of limitations such as pixel crosstalk and finite spatial resolution. This is discussed in more detail in the Supplementary Information (Supplementary Figs. 11–15). For the most complicated spatiospectral states that require all 90 spatial and polarisation modes to be excited equally across the entire addressable frequency band, the proportion of the total output power delivered to the target state is typically 86% where the spectral features are wider than 30 GHz ($90 \times (4400/30) = 13{,}200$ spatiospectral modes). This value drops to 77% for 20 GHz features (19,800 spatiospectral modes) and 39% for 10 GHz features (39,600 spatiospectral modes), which exceeds the spectral width of the beams on the SLM (~15 GHz).

We have demonstrated a system capable of measuring and generating arbitrary vector spatiospectral optical fields. This device is able to simultaneously control all classical linear degrees of freedom in an optical beam independently, enabling full time reversed optical waves to be generated through complex media, as well as spatiotemporal control of light more generally for applications such as imaging, nonlinear optics and micromanipulation. Alternate designs and options for scaling this device to higher number of spatial modes, longer delays, finer temporal features, and/or lower loss, are discussed in the Supplementary Information.

## Data availability

Data available on request from the authors.

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

## Acknowledgements

We acknowledge the Discovery (DP170101400, DE180100009) programme of the Australian Research Council, and the support of NVIDIA Corporation with the donation of the GPUs used for this research. We acknowledge Martin Plöschner for helpful discussions and Marcos Maestre Morote for work on the swept-wavelength trigger circuitry.

## Author contributions

Experiments performed by M.M. and N.F. with assistance from J.C.A.-V., R.R. and H.C. Optical system designed by N.F., D.N., M.M. and J.C. Spectral pulse shaper section designed by D.N. and N.F., MPLC by N.F. and J.C., spatiotemporal holograms designed by M.M., N.F. and J.C. Optical system assembled by N.F. Data analysis by M.M., N.F. and J.C. Manuscript written by J.C., M.M. and N.F. with input from all authors. J.C. conceived and supervised the project.

## Competing interests

The authors declare no competing interests.
