## [Peer Review File · Nature Communications]

REVIEWER COMMENTS

Reviewer #1 (Remarks to the Author):

The paper reports on optical system and methods for spatial temporal and polarisation control of light propagating through a complex medium. Any light field distribution across all these dimensions is possible to synthesize within the available space-time and its frequencies. With no doubt these are outstanding results well worth publishing in highly exposed media as they open numerous important applications in e.g. telecommunications and advanced imaging. The paper rigorous, thorough, clearly written with exemplary graphics and numerous additional studies compiled in supplementary information. The explanatory media provided for simpler understanding are simply outstanding. I haven't spotted any inaccuracies or unjustified conclusions.

My suggestions, which should all be considered optional, would only aim at the presentation. The paper invests rather large space on justifying its novelty, which I think is not necessary. Similarly, fig 3 is maybe excessive in including all the demonstrations. At the same time the main text does not feature any quantitative assessments so I would recommend including the most important studies, currently hidden in supplements, particularly those regarding the fidelity and desired field purity in the main text since it would be of interest for the readers before reaching the conclusions.

Reviewer #2 (Remarks to the Author):

The authors demonstrate the control of light-fields in a way that I have not seen before in related works. The authors explain, in a very educational way, the problem of controlling polarisation + 3spatiotemporal dimensions. Afterwards, they explain how to exploit their recent breakthrough technology (MPLC) to overcome this challenge.

They show how to control the $x,y,z/t$ coordinate of both the horizontal and vertical component of EM-fields independently, and remarkably demonstrate this technology in Fig3.

The article is very well written, the results are impressive, and to the best of my knowledge new. Both the abstract video (the short one) and the 1h long details will help fellow researchers to adapt their technology. I very much enjoyed these videos. They detail their work in such an excellent visual way too.

Their new technology will undoubtedly be employed and adapted by fellow researchers, to observe new phenomena both in the classical and quantum world. Many theoretical ideas would be exciting to see in lab implementations which have been impossible until now. One example that comes to mind immediately is the fundamental 3-dimensional uncertainty of a photon wavefunction. It requires the control of the three spatial dimensions plus polarisation (as it requires focusing on the non-paraxial limit). This new technology will enable many other fundamental studies, I think.

There is only one small suggestion/question: The authors do not mention the applicability to the single-photon/quantum regime. I believe it should be straightforwardly possible, but a statement from the authors in the manuscript might be interesting.

Of course, I recommend the acceptance of this article in Nature Communication.

Reviewer #3 (Remarks to the Author):

This manuscript describes an optical spatial-spectral shaper that provides for arbitrary space-spectral (equivalent to space-time) field transformations in two independent polarizations. (When we say arbitrary

we should understand this means subject to certain constraints, such as spatial and spectral resolution limits and instrumental imperfections.) Both optical spectral-temporal shaping and 2D spatial shaping individually have an extensive history. There are also a number of works that report spectral-temporal plus 1D spatial shaping, as well as the use of such techniques with highly scattering media including multimode fibers. However, the shaping in the current work, which achieves arbitrary spectral-temporal control in 2D spatially as well as two polarizations by combining pulse shaping with mode sorting technology (developed by authors for multimode fiber transmission) is unique, to my knowledge. This allows generation of many unique waveforms 3D waveforms (viewing time as equivalent to the longitudinal or z dimension). This is an exciting development.

Prior to publication, however, the manuscript needs extensive work.

From a mechanical perspective, the manuscript has many, many instances of sentence fragments. For example, in first paragraph on page 3: "For example, the control of light propagation through complex media for applications such as imaging. Analogous to previous demonstrations for lower frequency phenomena." Neither of these is a complete sentence. This violation of basic grammar permeates the manuscript. Authors must correct this repeated, basic grammar violation throughout the manuscript, both in the main text and in the supplementary information.

There is substantial detail in the supplementary information document. However, I am disappointed at the lack of substantial detail in the main text, which reads more like a public relations blurb than a scientific paper. I recognize that there is not room for all the detail in the main text. But authors need to do a better job of bringing in enough detail so that a reader basically familiar with this area can feel somewhat satisfied without having to take the time to wade through the long supplementary document. I will give several examples in the following bullets:

1. The experimental setup needs to be better described. For example, the comment that the output of the spectral-spatial shaper is a 1D array of 45 Gaussian spots is at best confusing. Apparently the mode sorter requires an array of input spots with certain size and spacing for proper operation. However, the spectral-spatial shaper outputs into free-space and is not intrinsically constrained to generate output spots at certain locations. It could be programmed to produce very different spatial outputs. However to be useful in the overall setup, it should be programmed to generate spots of desired size on the 1×45 grid. This could be explained in a way that is understandable – but in current writing it is only clear after poring through the supplementary information and the reference on the mode sorter.
2. The multimode fiber needs better specification, e.g., length, differential group delay, some estimate of number of spatial modes supported.
3. Page 3, 2nd column: "This linear description consists of a set of 90×90 frequency-dependent complex matrices through which any spatiotemporal input can be mapped to any spatiotemporal output in both directions." Please explain where the 90×90 comes from. (I figured it out when I spent time on the supplementary information, but was puzzled when I read the main text. Another indication the narrative in the main text needs to be improved.)
4. Renderings in Fig. 3 are attractive but need a better explanation. For the plots on black background, it is a 2D plot – how are they portraying all of x, y , and t in a 2D plot? And what is the point of the amplitude-phase color plot at the bottom right? Probably there are simple explanations, but authors should take the care to explain their portrayal so the reader doesn't have to guess.
5. Aside from providing pretty images, is there any way to quantify or characterize the fidelity of the mode transforms achieved in experiment? Nothing is presented in the main text, which in my opinion detracts from the scientific quality. Once again, there is quite a bit in the supplemental information – why don't authors draw some appropriate conclusions as to the fidelity in the main text, then refer to specific parts of supplementary information for backup? A few things that jumped out to me while skimming through the supplementary information include: (1) According to Fig. S9, the average impulse response seems to be

about 2 ps, which is about 5 times wider than one would expect with the specified 4.4 THz spectrum. This seems to be a significant broadening, i.e., loss of temporal resolution. (2) Also from Fig. S9, the transmission at plus or minus 2 psec seems to be down by about 50% compared to transmission at zero delay. Why? This seems to imply a significant loss of spectral resolution.

As explicitly stated by authors, all the spectral/temporal properties are measured in the frequency domain using a swept wavelength source; temporal information is obtained via Fourier transform. Since there is never a short pulse input, authors are not correct to claim they have generated the claimed space-time fields. What they have is a measurement that predicts what space-time field their apparatus would produce if illuminated with a specified ultrashort pulse input. There is no reason to expect that this prediction will be wrong. Still it is not the same as claiming they have actually made the described space-time fields, which they have not. There may be significant additional challenge in experiments with ultrashort pulses, such as performing the waveform characterization (the trick of making all the measurements with a wavelength tunable laser no longer works, now one has to come up with some new trick to measure the actual 3D shaped field).

For the most part, the referencing is good. But I would like to point authors' attention to the field of spectral-spatial holography using spectral hole burning materials. This approach is also capable in principle of 3D waveform generation (time plus 2D in space). Authors should check what experimental results have been achieved and reference accordingly. I would also like to point out several additional references, some of which may be worth including. (a) regarding space-time focusing and time reversal in microwave wireless: Dezfoolijan et al, *Opt. Lett.* 38, 4946 (2013); *IET Communications* 7, 1287 (2013); *IEEE Wireless Communications* 1, 520 (2012). (b) regarding space-time focusing in a multimode fiber: Liu et al, *Opt. Lett.* 43, 4675 (2018). (c) regarding spectral-polarization shaping for polarization mode dispersion compensation or emulation: Miao et al, *Opt. Lett.* 32, 2360 (2007); *IEEE Photonics Technology Letters* 20, 159 (2008).

Regarding the Supplementary Information (SI):

There is a great deal of useful information in this document. However, as stated above, I strongly suggest to summarize in the main text some of the key results that are detailed the SI. Furthermore, do a better job in the main text of referring the reader to specific sections in the SI for specific more detailed information (the main points of which should be stated in the main text).

Authors should also fix the sentence fragment problem that they have through the document. And go through to correct any obvious errors, such as that which I found immediately on p. 3: "This consists of polarization diversity optics (green in Fig. S1), dispersive and beam resizing optics (red), and the spatial light modulator (SLM) (blue)." As far as I can tell, the colors listed are mixed up.

Reviewer #1 (Remarks to the Author):

The paper reports on optical system and methods for spatial temporal and polarisation control of light propagating through a complex medium. Any light field distribution across all these dimensions is possible to synthesize within the available space-time and its frequencies. With no doubt these are outstanding results well worth publishing in highly exposed media as they open numerous important applications in e.g. telecommunications and advanced imaging. The paper rigorous, thorough, clearly written with exemplary graphics and numerous additional studies compiled in supplementary information. The explanatory media provided for simpler understanding are simply outstanding. I haven't spotted any inaccuracies or unjustified conclusions.

My suggestions, which should all be considered optional, would only aim at the presentation. The paper invests rather large space on justifying its novelty, which I think is not necessary. Similarly, fig 3 is maybe excessive in including all the demonstrations. At the same time the main text does not feature any quantitative assessments so I would recommend including the most important studies, currently hidden in supplements, particularly those regarding the fidelity and desired field purity in the main text since it would be of interest for the readers before reaching the conclusions.

Quantitative numbers for fidelity in the Main Document were lacking, with most of those metrics previously featuring only in the Supplementary Information. We've now summarised them in the Main Document as well (final ~1/3 of a page), as well as added a new page and Fig. S17 in the Supplementary Information, which looks at the distribution in spatial fidelity degradation due to mode dependent loss in the system.

Although the introduction could be considered relatively long, given that we have the space in the Nature Comms. format, we'd like to leave it in. For an expert, well-versed in the field, it would be unnecessarily long, but we think it's also important to cater for non-experts and early PhD students. Similarly, the video has about 15 minutes at the start which is more tutorial or review paper style, before moving into the actual results themselves. The goal is to bring the reader up to speed from a lower level of expertise in this specific area.

Reviewer #2 (Remarks to the Author):

The authors demonstrate the control of light-fields in a way that I have not seen before in related works. The authors explain, in a very educational way, the problem of controlling polarisation + 3spatiotemporal dimensions. Afterwards, they explain how to exploit their recent breakthrough technology (MPLC) to overcome this challenge.

They show how to control the x,y,z/t coordinate of both the horizontal and vertical component of EM-fields independently, and remarkably demonstrate this technology in Fig3.

The article is very well written, the results are impressive, and to the best of my knowledge new. Both the abstract video (the short one) and the 1h long details will help fellow researchers to adapt their technology. I very much enjoyed these videos. They detail their work in such an excellent visual way too.

Their new technology will undoubtedly be employed and adapted by fellow researchers, to observe new phenomena both in the classical and quantum world. Many theoretical ideas would be exciting to see in lab implementations which have been impossible until now. One example that comes to mind immediately is the fundamental 3-dimensional uncertainty of a photon wavefunction. It requires the control of the three spatial dimensions plus polarisation (as it requires focusing on the non-paraxial limit). This new technology will enable many other fundamental studies, I think.

There is only one small suggestion/question: The authors do not mention the applicability to the single-photon/quantum regime. I believe it should be straightforwardly possible, but a statement from the authors in the manuscript might be interesting.

Of course, I recommend the acceptance of this article in Nature Communication.

We've added a reference to two-photon interference in multimode optical fibre as an example from quantum mechanics. We now include quantum optics as a potential application when listing off use cases. "Applications such as nonlinear microscopy, micromachining, quantum optics, ..."

Reviewer #3 (Remarks to the Author):

This manuscript describes an optical spatial-spectral shaper that provides for arbitrary space-spectral (equivalent to space-time) field transformations in two independent polarizations. (When we say arbitrary we should understand this means subject to certain constraints, such as spatial and spectral resolution limits and instrumental imperfections.) Both optical spectral-temporal shaping and 2D spatial shaping individually have an extensive history. There are also a number of works that report spectral-temporal plus 1D spatial shaping, as well as the use of such techniques with highly scattering media including multimode fibers. However, the shaping in the current work, which achieves arbitrary spectral-temporal control in 2D spatially as well as two polarizations by combining pulse shaping with mode sorting technology (developed by authors for multimode fiber transmission) is unique, to my knowledge. This allows generation of many unique waveforms 3D waveforms (viewing time as equivalent to the longitudinal or z dimension). This is an exciting development.

Prior to publication, however, the manuscript needs extensive work.

From a mechanical perspective, the manuscript has many, many instances of sentence fragments. For example, in first paragraph on page 3: “For example, the control of light propagation through complex media for applications such as imaging. Analogous to previous demonstrations for lower frequency phenomena.” Neither of these is a complete sentence. This violation of basic grammar permeates the manuscript. Authors must correct this repeated, basic grammar violation throughout the manuscript, both in the main text and in the supplementary information.

We found 4 sentence fragments in the Main Document, and some more in the Supplementary Information (mostly in the FAQ section, which was written in a conversational style). These sentences have been adjusted.

There is substantial detail in the supplementary information document. However, I am disappointed at the lack of substantial detail in the main text, which reads more like a public relations blurb than a scientific paper. I recognize that there is not room for all the detail in the main text. But authors need to do a better job of bringing in enough detail so that a reader basically familiar with this area can feel somewhat satisfied without having to take the time to wade through the long supplementary document. I will give several examples in the following bullets:

1. The experimental setup needs to be better described. For example, the comment that the output of the spectral-spatial shaper is a 1D array of 45 Gaussian spots is at best confusing. Apparently the mode sorter requires an array of input spots with certain size and spacing for proper operation. However, the spectral-spatial shaper outputs into free-space and is not intrinsically constrained to generate output spots at certain locations. It could be programmed to produce very different spatial outputs. However to be useful in the overall setup, it should be programmed to generate spots of desired size on the 1×45 grid. This could be explained in a way that is understandable – but in current writing it is only clear after poring through the supplementary information and the reference on the mode sorter.

The system is explained at a few levels of detail. Almost purely conceptual in Fig. 1, with additional detail in Fig. 2. 3D rendered schematic in Fig. S2 and photograph in Fig. S3/S4. We’ve added an extra sentence to emphasise that selecting a 1D spot position using the spectral pulse shaper subsystem, excites a specific HG mode from a 2D set.

The operation of the spectral pulse shaper sub-system is similar to [35] or a traditional wavelength selective switch. Ensuring you generate the correct spots at the correct locations is part of the calibration procedure when building the device, as well as the hologram calculation.

As you mention, rather than generating spot arrays, it would be possible to implement some additional spatial shaping of the beams, at least spatially in 1D, using the SLM. That would be similar to approaches such as [11],[32],[33], with the associated limitations. We use the mode sorter here, and the relatively new 1D-to-2D spatial transformation it implements, to allow the spectral pulse shaper sub-system to select HG modes in 2D from a kind of 1D “lookup table”. In addition to decreasing the number of pixels required of the SLM (as some of the beam shaping is being performed by the fixed masks of the MPLC), it allows two spatial dimensions to be packed onto a single dimension of the SLM.

Our spatiotemporal beam shaping system is inherently a three-dimensional device, that uses relatively new technology like the 1D-to-2D MPLC transform. This does make it a bit confusing to understand when attempting to summarise the device with text and some 2D flattened diagrams. However, as highlighted by the other Reviewers, this is really where the Supplemental Videos are incredibly useful. In the ~6 minute video abstract, designed for the casually interested, the optical path is briefly summarised from 3:25 to 5:00. In the full technical video, the fly through of the optical path is covered in detail from 14:00 to 21:00. It means the reader does not need to try to piece together the device in their head from a collection of 2D figures and text. There is an animation which walks them through every element in 3D. This really helps with clarity of understanding, as we can show the reader directly how the spatial and temporal/spectral dimensions are handled, and how the beam transforms between 2D and 3D forms. This work is discussing the manipulation of light in 3D, there is significant benefit from being able to explain the principles of operation in this way.

2. The multimode fiber needs better specification, e.g., length, differential group delay, some estimate of number of spatial modes supported.

We’ve added the line “The fibre supports the same number of modes as the spatiotemporal beam shaper (90 spatial/polarisation modes) with a delay spread of approximately 0.15 ps/m.” The length was stated in the main document (5 m length of graded-index 50 μm core diameter), with a citation to [52] which outlines the fibre in detail, and discussed at 26:05 and 56:18 in the video. The fibre supports the same number of modes as the pulse shaper device itself (45 spatial modes per polarisation).

3. Page 3, 2nd column: “This linear description consists of a set of 90×90 frequency-dependent complex matrices through which any spatiotemporal input can be mapped to any spatiotemporal output in both directions.” Please explain where the 90×90 comes from. (I figured it out when I spent time on the supplementary information, but was puzzled when I read the main text. Another indication the narrative in the main text needs to be improved.)

The significance of 90 should now be clearer from the sentence added above. We’ve also added an extra sentence immediately after the 90×90 frequency-dependent complex matrices are mentioned.

4. Renderings in Fig. 3 are attractive but need a better explanation. For the plots on black background, it is a 2D plot – how are they portraying all of x,y, and t in a 2D plot? And what is the point of the amplitude-phase color plot at the bottom right? Probably there are simple explanations, but authors should take the care to explain their portrayal so the reader doesn’t have to guess.

The plots display 3D information as a sequence of 2D slices (left hand side of Fig. 3), as well as a volumetric plot on the right-hand side to help the reader visualise the 3D field. In the Supplemental

Videos, these are shown as 3D volumetric renderings with changing perspective in playback time, as well as animated 2D slices.

The amplitude/phase colourmap is necessary because the plots are showing the optical field, where light-dark represents amplitude and colour represents phase. For examples such as Fig. 3c, it illustrates that the eyes and mouth of the smiley face is out of phase with respect to the head outline. For Fig. 3f, it indicates the phase relationship between the polarisations required to generate the radially polarised clock hand. In the Supplementary Video, there are also various examples such as the generation of LG modes, where the spatial phase information is important.

We've added an extra two sentences. "The measured 3D optical fields (2D space and 1D time/frequency dimension) are plotted as a sequence of 2D fields as well as volumetric renderings. The plots contain both amplitude and phase information for these fields for both polarisation components." We've also added an extra annotation to Fig. 3 that says 'Field colourmap' directly above the colourmap.

5. Aside from providing pretty images, is there any way to quantify or characterize the fidelity of the mode transforms achieved in experiment? Nothing is presented in the main text, which in my opinion detracts from the scientific quality. Once again, there is quite a bit in the supplemental information – why don't authors draw some appropriate conclusions as to the fidelity in the main text, then refer to specific parts of supplementary information for backup? A few things that jumped out to me while skimming through the supplementary information include:

You're correct we should have included a summary of the fidelity in the main text. We've added about a third of a page of extra discussion at the end of the paper, which provides a breakdown of the different mechanisms which contribute the imperfections. As well as some additional details related to the spatiotemporal focus example that was previously only in the text associated with Fig. S16.

We've also included an extra page, including the new Fig. S17 which goes into some depth regarding the way mode dependent losses contribute to a degradation in the fidelity of the output spatial states.

(1) According to Fig. S9, the average impulse response seems to be about 2 ps, which is about 5 times wider than one would expect with the specified 4.4 THz spectrum. This seems to be a significant broadening, i.e, loss of temporal resolution.

This is just a misreading of what Fig. S9 is displaying. Fig. S9 is the raw impulse response of the system as a whole, it's not a demonstration of any temporal focusing. It is the impulse response of the MMF, plus some delay inside the spectral pulse shaper device itself. We think you're interpreting Fig. S9 more like the examples of Fig. 3b, Fig. 3c, Fig. 3e, Fig. S16. Those demonstrate the ability to deliver a specific spatial pattern to a specific single time slot. In those tests you can see that almost all the power is delivered to the allocated time slot, with little broadening into adjacent delays within the bandwidth limits of the device.

However it was a bit confusing given the way we organised the figures. We had 3 sub-plots in Fig. S9, which made for efficient page space usage, but conceptually, those 3 sub-plots are not all linked. So we split the sections up. Now 'Characterisation of impulse response' and 'Characterisation of delay-dependent loss' have been separated into different chunks of text, each with their own Figures (S9 and S10) respectively.

(2) Also from Fig. S9, the transmission at plus or minus 2 psec seems to be down by about 50% compared to transmission at zero delay. Why? This seems to imply a significant loss of spectral resolution.

This is just a misinterpretation of what the graph is illustrating. There's no significant loss difference between 0ps and 2ps, the ~50% transmission difference is simply because of the way the test is defined. For the 0 ps example, all the light is delivery to a single delay (H and V both arrive at 0 ps), rather than being split amongst two delays (H arrives at 0 ps and V arrives at x ps). So it is naturally going to be twice as intense at 0ps delay, as all the light arrives in a single time slot. When light is split up amongst two pulses, as it is for the other delays, the intensity it halved. An alternative test would be to generate a single spot at a single variable delay. We decided to do a slightly more difficult test where the device needs to address a span of delay, rather than a single delay. This demonstration is explained in the text above Fig. S9, as well in Fig. S10, and discussed in the video at 55:40.

As explicitly stated by authors, all the spectral/temporal properties are measured in the frequency domain using a swept wavelength source; temporal information is obtained via Fourier transform. Since there is never a short pulse input, authors are not correct to claim they have generated the claimed space-time fields. What they have is a measurement that predicts what space-time field their apparatus would produce if illuminated with a specified ultrashort pulse input. There is no reason to expect that this prediction will be wrong. Still it is not the same as claiming they have actually made the described space-time fields, which they have not. There may be significant additional challenge in experiments with ultrashort pulses, such as performing the waveform characterization (the trick of making all the measurements with a wavelength tunable laser no longer works, now one has to come up with some new trick to measure the actual 3D shaped field).

In this work we are not trying to prove that time/frequency domain representations are equivalent, or that space-time fields exist. There's nothing new or surprising about that aspect, that's a given. Rather we are presenting the first device capable of generating space-time /space-spectral non-separable beams, and characterising the device's performance. Hence our goal here is really to characterise the performance of the device as accurately as possible, which is why we use the frequency-domain. For other researchers interested in the paper, presumably the information of interest is; "what did they build?" and "how well does it work?". Our frequency-domain approach is what has allowed us to provide such detailed information, particularly in the Supplementary Information, regarding how the device performs. Ultimately, as the device is a frequency-domain device, it makes sense to characterise it in that domain.

As discussed in the Frequently Asked Questions section of the Supplementary Information, we could have instead characterised in the time-domain rather than the frequency-domain. However, the important question is, what would we learn from a time-domain characterisation? In practice, time-domain characterisation would require building a separate time-domain characterisation apparatus, based on a pulsed-source and delay line. Which in practice would then be calibrated against the existing frequency-domain characterisation apparatus acting as the "gold standard". The experiments would then be redone in the time-domain. However because the time and frequency domain characterisations must be consistent, any measured inconsistency between the two would be a measure of the inaccuracies in the new time-domain characterisation apparatus, rather than the device under test. That is, we'd end up with a characterisation of the new time-domain apparatus itself, not the device-under-test.

It's a bit like if you measure a test mass on a well calibrated and trusted scale, and then measure it again on a more poorly calibrated scale. The second measurement isn't a measure of your test mass, it's a measure of the second set of scales.

It's similar to why many microwave system are often characterised using a VNA in the frequency domain, even if the property of interest is in the time-domain. Frequency-domain gets you better dynamic range, often more bandwidth, and importantly, better calibration (as calibration is frequency-dependent, not time-dependent). We measure the performance of our device using the most accurate experimental method available, which for this wavelength band and for our laboratory, is a swept-wavelength digital holography characterisation apparatus. This is regularly used for characterising the spatiotemporal properties of multimode fibre devices for telecommunications.

For the most part, the referencing is good. But I would like to point authors' attention to the field of spectral-spatial holography using spectral hole burning materials. This approach is also capable in principle of 3D waveform generation (time plus 2D in space). Authors should check what experimental results have been achieved and reference accordingly. I would also like to point out several additional references, some of which may be worth including.

We've added a reference to time-space holography by spectral hole burning materials

<https://doi.org/10.1364/JOSAB.3.000527>

(a) regarding space-time focusing and time reversal in microwave wireless:

Dezfooliyan et al, Opt. Lett. 38, 4946 (2013); <https://doi.org/10.1364/OL.38.004946>

IET Communications 7, 1287 (2013); <https://doi.org/10.1049/iet-com.2012.0768> {Citation added}

IEEE Wireless Communications 1, 520 (2012).

(b) regarding space-time focusing in a multimode fiber: Liu et al, Opt. Lett. 43, 4675 (2018).

<https://doi.org/10.1364/OL.43.004675> {Citation added}

(c) regarding spectral-polarization shaping for polarization mode dispersion compensation or emulation: Miao et al, Opt. Lett. 32, 2360 (2007); <https://doi.org/10.1364/OL.32.002360>

IEEE Photonics Technology Letters 20, 159 (2008). <https://doi.org/10.1109/LPT.2007.912494> {Citation added}

Regarding the Supplementary Information (SI):

There is a great deal of useful information in this document. However, as stated above, I strongly suggest to summarize in the main text some of the key results that are detailed the SI. Furthermore, do a better a job in the main text of referring the reader to specific sections in the SI for specific more detailed information (the main points of which should be stated in the main text).

Hopefully this has been clarified with the changes mentioned above or in the video.

Authors should also fix the sentence fragment problem that they have through the document. And go through to correct any obvious errors, such as that which I found immediately on p. 3: "This consists of polarization diversity optics (green in Fig. S1), dispersive and beam resizing optics (red), and the spatial light modulator (SLM) (blue)." As far as I can tell, the colors listed are mixed up.

Indeed, the colours named in the text were from an earlier draft of the figure, this has been updated.

REVIEWERS' COMMENTS

Reviewer #1 (Remarks to the Author):

The new text is an improved version of already great paper, my comments were fully addressed and I see no further obstructions standing in the way of publishing the paper.

Reviewer #3 (Remarks to the Author):

I have read through the author's response and the revised main article. I did not look at the revised supplementary information.

I do find the article substantively improved in many aspects, though I do have a few remaining criticisms (see below). The description of the experimental setup in the main text is sufficiently improved, the references I have suggested are added, and extra text is introduced in the main text speaking to the fidelity of the results achieved. This new text in particular is a major improvement and does a nice and appropriate job of summarizing key points concerning fidelity and experimental limitations, with substantial additional detail provided in the Supplementary Information.

As I already mentioned in original review, the work does represent an exciting new development which is indeed at the level appropriate for publication in Nature Communications.

With respect to remaining criticisms, I have the following that I believe should be addressed.

(1) The first concerns the sentence fragment issue. Authors' response letter states: "We found 4 sentence fragments in the Main Document, and some more in the Supplementary Information (mostly in the FAQ section, which was written in a conversational style). These sentences have been adjusted."

I looked again at the original manuscript and found something like 17 fragments in the main article. I have highlighted these and will attach with my review. In the revised main article, in addition to many original fragments that were not corrected, I noted three instances where text in red (new or modified text) constitute sentence fragments. I have also highlighted these on the revised main text and have attached the file (these are not all the fragments, just the new ones).

I did not and will not recheck the Supplementary Information. Again editor can decide if this is an issue for this journal or not.

(2) I originally commented: "As explicitly stated by authors, all the spectral/temporal properties are measured in the frequency domain using a swept wavelength source; temporal information is obtained via Fourier transform. Since there is never a short pulse input, authors are not correct to claim they have generated the claimed space-time fields. What they have is a measurement that predicts what space-time field their apparatus would produce if illuminated with a specified ultrashort pulse input. There is no reason to expect that this prediction will be wrong. Still it is not the same as claiming they have actually made the described space-time fields, which they have not. There may be significant additional challenge in experiments with ultrashort pulses, such as performing the waveform characterization (the trick of making all the measurements with a wavelength tunable laser no longer works, now one has to come up with some new trick to measure the actual 3D shaped field)."

Authors respond: "In this work we are not trying to prove that time/frequency domain representations are equivalent, or that space-time fields exist. There's nothing new or surprising about that aspect, that's a given. Rather we are presenting the first device capable of generating space-time /space-spectral non-separable beams, and characterising the device's performance. Hence our goal here is really to characterise the performance of the device as accurately as possible, which is why we use the frequency-domain. For other researchers interested in the paper, presumably the information of interest is; "what did they build?"

and “how well does it work?”. Our frequency-domain approach is what has allowed us to provide such detailed information, particularly in the Supplementary Information, regarding how the device performs. Ultimately, as the device is a frequency-domain device, it makes sense to characterise it in that domain. ... As discussed in the Frequently Asked Questions section of the Supplementary Information, we could have instead characterised in the time-domain rather than the frequency-domain. However, the important question is, what would we learn from a time-domain characterisation? In practice, time-domain characterisation would require building a separate time-domain characterisation apparatus, based on a pulsed-source and delay line. Which in practice would then be calibrated against the existing frequency-domain characterisation apparatus acting as the “gold standard”. The experiments would then be redone in the time-domain. However because the time and frequency domain characterisations must be consistent, any measured inconsistency between the two would be a measure of the inaccuracies in the new time-domain characterisation apparatus, rather than the device under test. That is, we’d end up with a characterisation of the new time-domain apparatus itself, not the device-under-test. ”

My new response: I do not dispute author’s argument. I agree author’s frequency domain characterization approach is effective and makes sense. That was not my point. My point is authors should be explicit that they have not yet actually created a 3D time-space-polarization signal; what they have done is create and characterize a system with controllable space-frequency (or space-time) transfer function that is additionally polarization multiplexed. Although this is expected to enable the claimed arbitrary space-time-polarization fields, they should acknowledge that to do so and measure the result will bring in additional (but not necessarily insurmountable) measurement complexity. I do feel that a few of authors’ statements in the manuscript are misleading in this regard. In particular: “Many of the beam types demonstrated here are being generated for the first time in any context” (immediately under Experimental Results) and “We have demonstrated a system capable of measuring and generating arbitrary vector spatiotemporal optical fields” (immediately under Conclusion). Neither is correct. Shaped space-time fields have not actually been generated, because there is never more than one input frequency at a time. Furthermore, the system is not yet “capable of measuring and generating arbitrary vector spatiotemporal optical fields,” because so far the system cannot measure multiple frequencies in a way that is sensitive to the spectral coherence of the input field. Perhaps the authors should also comment on what it would take to get to the actual demonstration using coherent ultrafast pulse inputs

Reviewer 3:

I do not dispute author's argument. I agree author's frequency domain characterization approach is effective and makes sense. That was not my point. My point is authors should be explicit that they have not yet actually created a 3D time-space-polarization signal; what they have done is create and characterize a system with controllable space-frequency (or space-time) transfer function that is additionally polarization multiplexed. Although this is expected to enable the claimed arbitrary space-time-polarization fields, they should acknowledge that to do so and measure the result will bring in additional (but not necessarily insurmountable) measurement complexity. I do feel that a few of authors' statements in the manuscript are misleading in this regard. In particular: "Many of the beam types demonstrated here are being generated for the first time in any context" (immediately under Experimental Results) *{Authors : This sentence has been removed}* and "We have demonstrated a system capable of measuring and generating arbitrary vector spatiotemporal optical fields" *{Authors : This has been changed to spatio-spectral optical fields}* (immediately under Conclusion). Neither is correct. Shaped space-time fields have not actually been generated, because there is never more than one input frequency at a time. Furthermore, the system is not yet "capable of measuring and generating arbitrary vector spatiotemporal optical fields," because so far the system cannot measure multiple frequencies in a way that is sensitive to the spectral coherence of the input field. Perhaps the authors should also comment on what it would take to get to the actual demonstration using coherent ultrafast pulse inputs.

We do not necessarily need spectral coherence to perform a spatio-spectral control (without temporal control) as in our Fig. 3a. However, in order to generate the spatiotemporal fields presented in this work (Fig. 3b to Fig. 3h), we do have spectral coherence. Specifically, we control the phase between each single wavelength to enable light delivery at a single time step (as in Fig. 3b and Fig. 3c), or over a short temporal interval (as in Fig. 3d to Fig. 3h). Hence, our work provides a spectrally coherent control of the input light (in addition to the spatial control), which is equivalent to spatiotemporal control, even though we have not physically launched a pulse of light.

We mentioned explicitly in the manuscript that the measurements are performed in the spectral domain: "All characterisation and results are measured in the frequency domain" and "For spatiotemporal beam demonstrations, the presented results are Fourier transformed into the time domain from the measured frequency dependent fields measured using swept-wavelength digital holography". Therefore, we believe that an additional section on characterisation techniques in the temporal domain is outside the scope of this work. We already discuss time vs. frequency domain in the 'Frequently Asked Questions' of the Supplementary Information, and in the online video.